# Satellite-Derived Annual Glacier Surface Flow Velocity Products for the European Alps, 2015–2021

Antoine Rabatel [1,*], Etienne Ducasse [1], Romain Millan [1] and Jérémie Mouginot [1,2]

1 Univ. Grenoble Alpes, CNRS, IRD, INRAE, Grenoble-INP, IGE (UMR 5001), 38000 Grenoble, France
2 Department of Earth System Science, University of California, Irvine, CA 92697, USA
* Correspondence: antoine.rabatel@univ-grenoble-alpes.fr; Tel.: +33-4-76-82-42-71

**Abstract:** Documenting glacier surface flow velocity from a longer-term perspective is highly relevant to evaluate the past and current state of glaciers worldwide. For this purpose, satellite data are widely used to obtain region-wide coverage of glacier velocity data. Well-established image correlation methods allow for the automated measurement of glacier surface displacements from satellite data (optical and radar) acquired at different dates. Although computationally expensive, image correlation is nowadays relatively simple to implement and allows two-dimensional displacement measurements. Here, we present a data set of annual glacier surface flow velocity maps at the European Alps scale, covering the period 2015–2021 at a 50 m × 50 m resolution. This data set has been quantified by applying the normalized cross-correlation approach on Sentinel-2 optical data. Parameters of the cross-correlation method (e.g., window size, sampling resolution) have been optimized, and the results have been validated by comparing them with in situ data on monitored glaciers showing an RMSE of 10 m/yr. These data can be used to evaluate glacier dynamics and its spatial and temporal evolution (e.g., quantify mass fluxes or calving) or can be used as an input for model calibration/validation or for the early detection of regional hazards associated with glacier destabilization.

**Dataset:** https://doi.org/10.57745/XHQ7TL

**Dataset License:** CC BY NC 4.0

**Keywords:** mountain glaciers; surface flow velocity; optical remote sensing; normalized cross correlation; Sentinel-2; European Alps

## 1. Summary

Glaciers flow thanks to gravity, transporting ice from the upper reaches, where accumulation processes dominate, to the lower elevations, where mass loss occurs by ablation and sometimes calving. An ice flow combines creeping, sliding (when the ice is temperate) and the deformation of underlying sediments (where these can be found). An ice flow therefore depends on the thermal regime of the glacier, the slope of the glacier bed, the glacier thickness and the water pressure at the ice/rock interface. For a glacier in dynamic equilibrium, the mass flux is balanced by surface processes, i.e., the mass transported to the glacier terminus is the same as the mass that is lost. In such conditions, a glacier does not change its geometry over time and is considered as "stable" (in a steady state) [1]. For advancing or retreating glaciers, the dynamic equilibrium is not maintained, and glacier flow velocities are higher or lower than required. Therefore, documenting the long-term evolution of glacier flow velocity reveals the past and current state of glaciers.

A way to measure glacier surface displacements is to follow structures that move with it (e.g., crevasses, debris . . . ) between two images acquired at different times. This was one of the first glaciological applications of Landsat images [2], at that time using visual analysis

and manual calculation. Since then, methods to derive glacier surface displacements from satellite data have been extensively explored: feature tracking on both optical and radar images and interferometry on radar images e.g., [3–5]. These methods were first applied on ice streams, draining two ice sheets: Greenland and Antarctica e.g., [6,7]. Indeed, such ice streams can be several kilometers long and present surface displacements of several meters per day, so they are thereby more adapted to the resolution of satellite data available at the end of the 20th century.

With the increase in spatial resolution and the temporal acquisition frequency of satellite images, the application of these methods become efficient for the much smaller glaciers in mountain regions around the world and their usually reduced surface displacements e.g., [8–11]. Near-global glacier surface flow velocity products started to be provided in the 2010s:

- GoLive (https://nsidc.org/data/golive, accessed on 1 March 2023), or the Global Land Ice Velocity Extraction from Landsat (GoLIVE) project, was created by the National Snow and Ice Data Center thanks to NASA funding. GoLIVE is a processing and staging system for near-real-time global ice velocity data derived from Landsat 8 panchromatic imagery applied to image pairs covering all glaciers > 5 km$^2$ and both ice sheets.
- ITS_LIVE (https://its-live.jpl.nasa.gov/, accessed on 1 March 2023), or the Intermission Time Series of Land Ice Velocity and Elevation (ITS_LIVE) project, part of NASA's MEaSUREs program, provides measurements of glacier and ice sheet surface velocity and elevation change at a high temporal resolution. The ITS_LIVE data product is a set of regional compilations of annual mean surface velocities for major glacier-covered regions, spanning the time period from 1985 to 2018, subject to image availability and quality, with a spatial resolution varying between 120 m × 120 m and 240 m × 240 m. Data scarcity and/or low radiometric quality are significant limiting factors for many regions in the earlier product years. Annual coverage is nearly complete for the years following the Landsat 8 launch in 2013.
- The data set "Alps glacier velocities 2013–2015" [12] (https://zenodo.org/record/3244871, accessed on 1 March 2023) contains the median glacier surface velocity for the Alps for the years 2013–2015 (Landsat 8). The velocities have been obtained by feature tracking Landsat images acquired 1 year apart.
- The data set "glacier surface velocities derived from Sentinel-1" [13] contains glacier surface velocity data for 12 glacierized regions on Earth, including the European Alps. Sentinel-1 radar data acquired within a time interval ranging from 6 to 48 days have been processed over the period January 2015 to January 2021. The velocity product is available at a 200 m × 200 m spatial resolution.

Regarding these products, the main weaknesses and gaps are as follows: (i) GoLIVE covers only those glaciers > 5 km$^2$ and has a low resolution (300 m). Only pairs of images with 16-day time intervals are treated, which is not appropriate for mountain glaciers [10,14]; (ii) ITS_LIVE does not cover the Alps; (iii) the product derived by Dehecq et al. [12] covers only the 2013–2015 period; and (iv) Sentinel-1-derived product, by [13], is limited for mountain glaciers owing to its coarse spatial resolution and because comprehensive velocity mapping can be hard to achieve for slow-moving glaciers such as in the Alps when using a short temporal cycle (i.e., 6–48 d).

Here, we present a data set of annually averaged glacier surface flow velocity products derived from Sentinel-2 optical data (i.e., since 2015), covering all glaciers in the European Alps (theoretical minimum size of 0.03 km$^2$ because of the correlation algorithm parameters) with a spatial resolution of 50 m × 50 m. To derive the glacier surface flow velocities, we used a normalized cross-correlation method with optimized parameter values [10–14] and a data reduction approach to filter the outputs and aggregate the individual measurements on a yearly basis [14]. This data set has been established as one of the satellite-derived products of the "Glacier Science in the Alps" project, named AlpGlacier, funded by the European Space Agency (ESA).

Such a data set and its public release is of potential benefit to better understand the spatial and temporal variability of glacier surface flow velocities in the current context of climate change (e.g., quantify mass fluxes, calving), as well as for modelers to invert the glacier surface velocity and retrieve the ice thickness distribution e.g., [11,15–17] or to calibrate and initiate ice flow models e.g., [18,19].

## 2. Data Description

The glacier surface flow velocity product was generated at the scale of the entire European Alps by using Sentinel-2 (ESA, Copernicus program) 10 m resolution optical images. Figure 1 shows the study region with the location of Sentinel-2 tiles that have been used in this study. Also shown in Figure 1 are the three areas, namely the Mont Blanc massif (France, Italy, Switzerland), the Zermatt region (Switzerland) and the Ötztal Alps (Austria), where Sentinel-2-derived surface flow velocity data were compared with in situ data for validation purposes by Mouginot et al. [14]. Indeed, in each of these areas, topographical measurements, made with d-GNSS, of the location of the ablation stakes used from glaciological observations are available. These data have an annual temporal resolution and decimeter precision. The d-GNSS measurements are point measurements, while the velocity maps obtained with Sentinel-2 data have a resolution of 50 m × 50 m. Thus, the comparison of the two is not perfect, and even if the differences are small enough for a fair comparison, it is possible that biases are still present. Therefore, Mouginot et al. [14] used these in situ data to evaluate the different filtering and aggregation methods used in the workflow of Sentinel-2 data processing to derive the annually aggregated glacier surface flow velocity maps. We refer the reader to Mouginot et al. [14] for a complete description of this validation step.

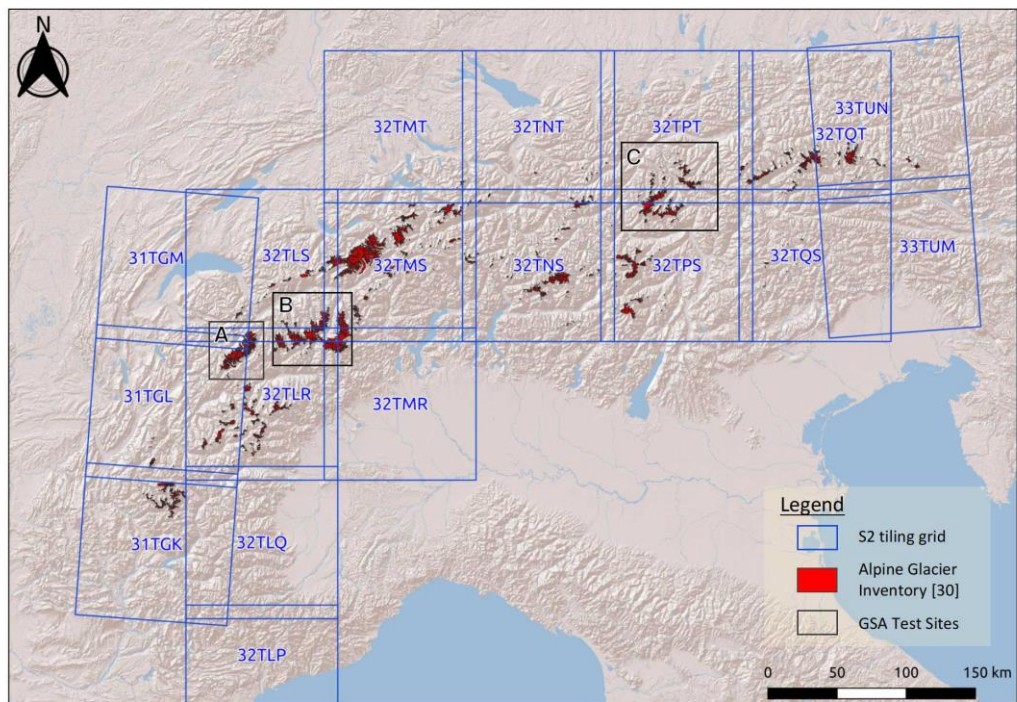

**Figure 1.** Sentinel-2 tiles covering the glaciers in the European Alps considered for glacier surface flow velocity mapping. Inset boxes show the three test sites where in situ point measurements of glacier surface velocity were available for validation purposes: (**A**) Mont Blanc massif; (**B**) Zermatt region; (**C**) Ötztal Alps.

For the quantification of the glacier surface flow velocity, Level 1C top of atmosphere reflectance data from Sentinel-2 A and B, with cloud cover lower than 80% have been used as input data. The Sentinel-2 satellite coverage encompasses 18 tiles over the European Alps (according to the UTM tiling grid proposed by ESA, Figure 1). In total, 1,211,867 pairs of images acquired between 30 August 2015 and 31 December 2021 have been processed.

### 2.1. Data Format and Content

The data presented in this manuscript are stored and shared in two different formats: netCDF (network common data form) and geoTIFF. The netCDF format consists, on the one hand, of a set of software libraries and, on the other hand, of a "self-documented" data format, independent of the hardware architecture, which allows the creation, access and sharing of scientific data stored in the form of tables. We follow the CF (climate and forecast) metadata conventions, which have been designed to promote the processing and sharing of files created with the netCDF API. The conventions define metadata that provide a definitive description of what the data in each variable represent and the spatial and temporal properties of the data. In the single netCDF file gathering all the glacier surface flow velocity products, the following architecture can be found:

- v2015–2016—annually aggregated glacier surface flow velocity for the hydrological year 2015–2016 (the hydrological year starts the 1st of October and finishes the 30 of September of the following year).
- v2016–2017—same as before but for the hydrological year 2016–2017.
- v2017–2018—same as before but for the hydrological year 2017–2018.
- v2018–2019—same as before but for the hydrological year 2018–2019.
- v2019–2020—same as before but for the hydrological year 2019–2020.
- v2020–2021—same as before but for the hydrological year 2020–2021.
- a—flow direction (= angle between the flow vector and $x$-axis, in radians).
- cnt—number of image pairs used to generate the displacement maps that have been aggregated to obtain the glacier surface flow velocity products (data per pixel).
- stdev—standard deviation on the velocity for each pixel.
- stdeva—standard deviation on the direction of glacier surface flow.
- flag—error flag mask on pixels considered as unreliable. Unreliable pixels—0; reliable pixels—1. Such pixels have been flagged according to the following criteria:
  - ○ Coefficient of variation of the glacier surface flow velocity higher than 75% to mask areas with unreliable variability (e.g., glacier edges).
  - ○ Standard deviation of the glacier flow direction higher than 2.5° to mask areas with unreliable flow direction changes.
- trend_2015–2021—trend on surface flow velocity over the study period (2015–2021).
- trend_mask—mask on pixels considered with a nonsignificant trend. Pixels showing time series with a nonsignificant trend (Kendall rank-of-correlation test with a $p$-value > 0.05) are masked. Masked pixels—0; nonmasked pixels—1.

Each layer of the netCDF file is also provided in a geoTIFF file (13 in total). The GeoTIFF (geographic tagged image file) format is a georeferenced raster file format. Data are provided in Universal Transverse Mercator, zone 32 north projection (EPSG: 32632). Figure 2 presents an example of the main layers that can be found in the netCDF file or as geoTIFF for the Mont Blanc area.

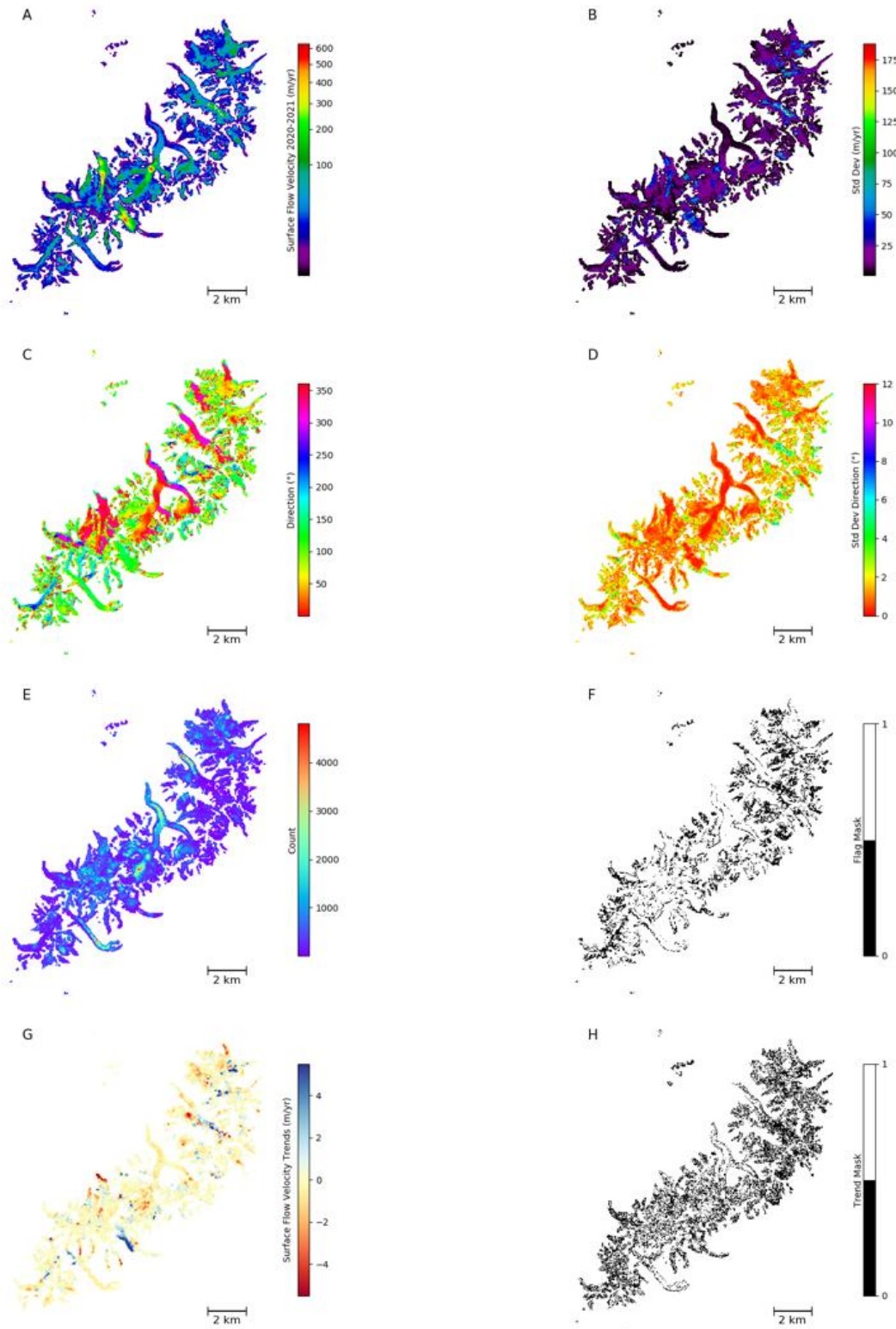

**Figure 2.** Examples of the data sets available in the netCDF file for the Mont Blanc test site. (**A**) Annually aggregated glacier surface flow velocity, (**B**) standard deviation of the glacier surface flow velocity, (**C**) glacier surface flow direction, (**D**) standard deviation of the glacier surface flow direction, (**E**) number of image pairs used for each pixel, (**F**) flag mask for pixels with unreliable values, (**G**) glacier surface flow velocity trends for the period 2015–2021, (**H**) mask for pixels with nonsignificant temporal trends.

Table 1 summarizes the main characteristics of our glacier surface flow velocity product and those of the other products available for the European Alps.

**Table 1.** Main characteristics of the glacier surface flow velocity products available for the European Alps.

| Product | Spatial Coverage | Spatial Resolution | Temporal Coverage | Temporal Resolution | Sensors |
|---|---|---|---|---|---|
| This study | European Alps | 50 m × 50 m | Since September 2015 | Annual average | Sentinel-2 |
| GoLIVE (https://nsidc.org/data/golive, accessed on 1 March 2023) | Global for glaciers > 5 km$^2$ | data | Since May 2013 | 16 days | Landsat-8 |
| "Alps glacier velocities 2013–2015" [12] | European Alps | 30 m × 30 m | 2013–2015 | Annual average | Landsat-8 |
| "Glacier surface velocities derived from Sentinel-1" [13] | 12 glacierized regions on Earth | 200 m × 200 m | January 2015–January 2021 | 6 to 48 days | Sentinel-1 |

*2.2. Overall Description, Strengths and Weaknesses*

The glacier surface flow velocity product covers the European Alps with void-free maps. Figure 3 shows examples of glacier surface flow velocity maps for the hydrological year 2020–2021 for three areas that contain a large variety of glaciers in terms of size, slope, thermal regime and therefore surface flow velocity patterns. Contrasted glacier velocities ranging from a few meters per year up to several hundreds of meters per year can be seen.

Very detailed patterns of flow velocity can be seen. For some areas around the median elevation and in the upper reaches of the glaciers (i.e., in the accumulation zone), some unreliable values persist (see below for the limitations). However, they can efficiently be flagged according to the different criteria that we have defined (e.g., coefficient of variation on the surface flow velocity and standard deviation on the surface flow direction) and masked.

Overall, the flow velocity of glaciers in the European Alps is rather low. Figure 4 illustrates the distribution of all 50 m × 50 m glacierized pixels covered by our product as a function of their annual flow velocity. One can see that 75% of the pixels have velocities lower than 25 m/yr and 50% lower than 10 m/yr. Therefore, with satellite data at a 10 m spatial resolution such as that from Sentinel-2 and because of the constrains of the temporal resolution of the acquired data, it remains a challenge to map the velocity of these small mountain glaciers. At the mountain-range scale, only annually aggregated values can be reliably derived.

Uncertainties can be found in the vicinity of the median elevation of glaciers (around 2900–3300 m a.s.l., depending on the orientation) and in the flat and often permanently snow-covered accumulation areas. Close to the median elevation, this is mainly because only a few images can be used for the correlation as the surface state at the end of summer quickly changes. In the flat accumulation areas, the surface is often homogeneous (no crevasses), which is also a challenge for the correlation. As mentioned earlier, a "flag mask," according to multiple criteria (e.g., coefficient of variation on the surface flow velocity and standard deviation on the surface flow direction), has been realized to handle these regions.

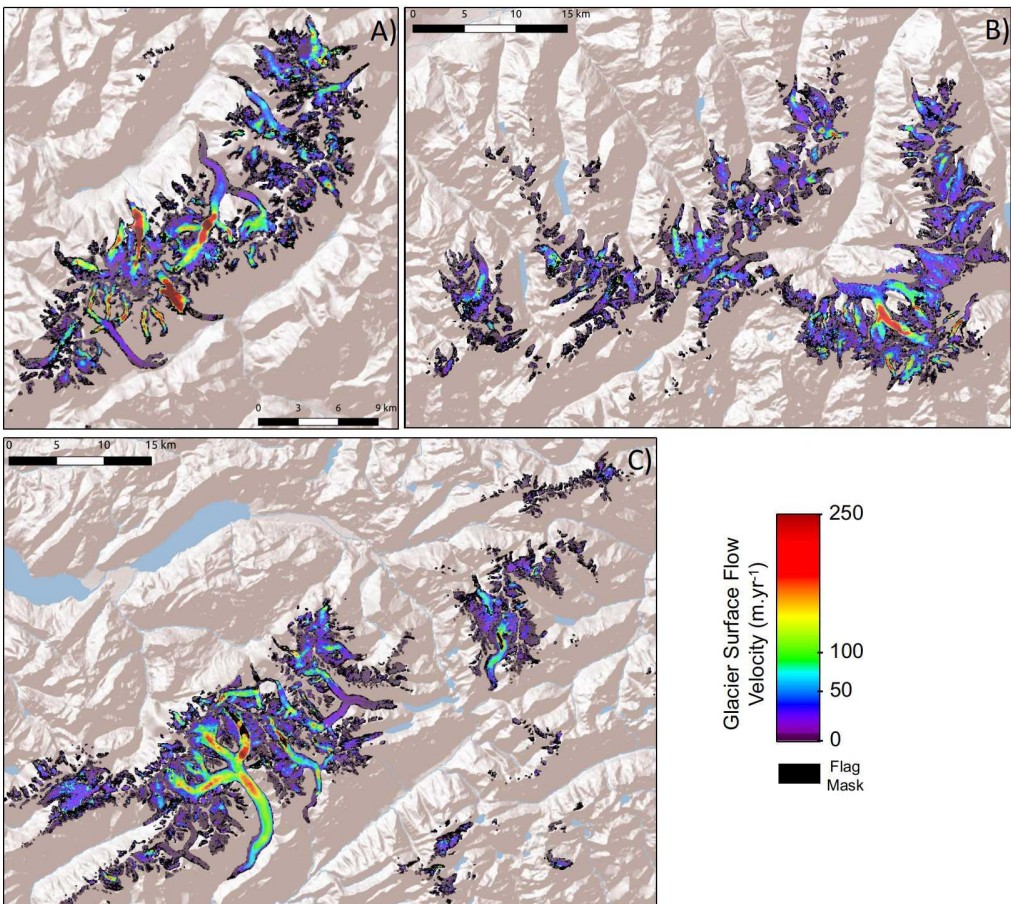

**Figure 3.** Example of glacier surface flow velocity maps for three areas in the European Alps ((**A**) Mont Blanc massif, (**B**) Zermatt area, (**C**) Upper Valais area with Aletsch Glacier, the largest in the European Alps). The glacier velocities are shown for the hydrological year 2020–2021.

Image pairs for the winter season are less numerous thanks to frequent cloud coverage and the lower performance of the correlation method because of snow coverage and extensive shadowing. Therefore, the annually aggregated glacier surface flow velocity maps are slightly biased by image pairs retrieved during the ablation period. Accordingly, if more short temporal offset image pairs give accurate displacement fields during summer, their weight in the annually aggregated map is lower than the one of long temporal offset pairs.

Moreover, the annual glacier surface flow velocity maps quantified over the period 2015–2021 are not based on the same amount of glacier surface displacement fields, because the number of available images has not been homogeneous. Indeed, the number of image pairs logically increased after the launch of Sentinel-2B, in 2017.

Figure 5 shows a comparison between our product and the only two other ones available at the European Alps scale and quantified from optical remote-sensing data. The other products use Landsat data processed with a chain similar to ours in terms of the parameters used for correlation. Our map better captures the main glaciers of the Mont Blanc massif, such as the Mer de Glace or Leschaux glaciers, with shear margin transitions that are better defined than in GoLIVE. Many small glaciers are not visible in the GoLIVE map, and the noise outside of the glaciers is important. In comparison with Dehecq et al. [12], the coverage of our mapping is most complete, which is due mainly to the restriction of this data set to image pairs with time interval of a year, whereas we compute all the possible image pairs between 5 and 400 days (see Section 3.2).

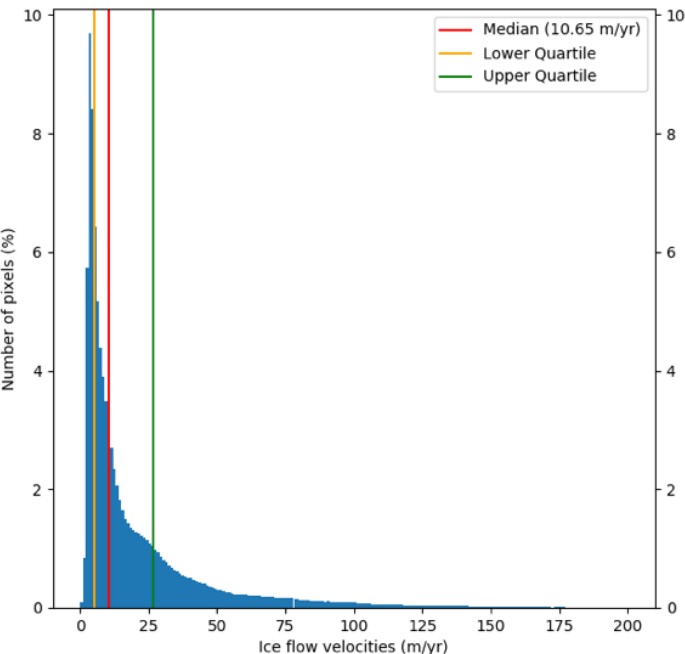

**Figure 4.** Distribution of the all 50 m × 50 m glacierized pixels in the glaciers covered by our product as a function of their annual surface flow velocity.

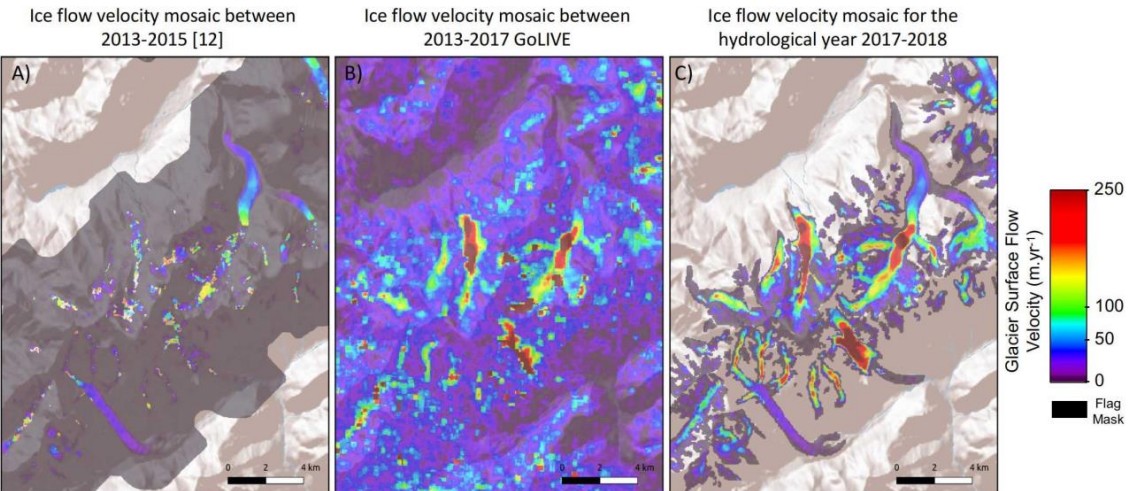

**Figure 5.** Comparison of glacier surface velocity maps obtained from the following: (**A**) adapted from [12] product; (**B**) mosaicking of GoLIVE products; (**C**) AlpGlacier product, from this study.

### 2.3. Overall Trends in Glacier Surface Flow Velocity

The trend over the entire study period (i.e., summer 2015–late 2021) can be quantified by using the time series of glacier surface displacements measured at different temporal offsets. An ordinary least squares (OLS) regression is performed, and the slope of the regression line is considered [14]. Figure 6 shows examples of OLS regressions (the red line in each plot, with the pale red envelope showing the uncertainty around the regression line) into the surface displacements' point clouds (blue dots with the vertical bar showing the uncertainty on the velocity and the horizontal bar showing the temporal range over which the velocity has been quantified). The three graphs illustrate different situations found at the Mont Blanc test site: (A) a location close to the terminus of Bossons Glacier showing a strong decrease in glacier surface flow velocity over the study period (around $-6$ m/yr$^2$); (B) a location on Brenva Glacier showing a strong increase (around $+9$ m/yr$^2$); and (C) a location on the tongue of Argentière Glacier where no significant trend can be encountered.

The trend is considered significant according to a Mann–Kendall test, and a mask has been generated on this basis. This test relies on the Kendall rank-of-correlation test [20,21]. It consists in a nonparametric test widely used in hydrology to assess the significance of a trend in a time series e.g., [22–24]. It is based on the measurement of the ordinal association between variables (in this case, ice flow surface velocities and date). The hypothesis of the presence of a monotonic trend in the time series is accepted with a *p*-value of the test that is below 0.05.

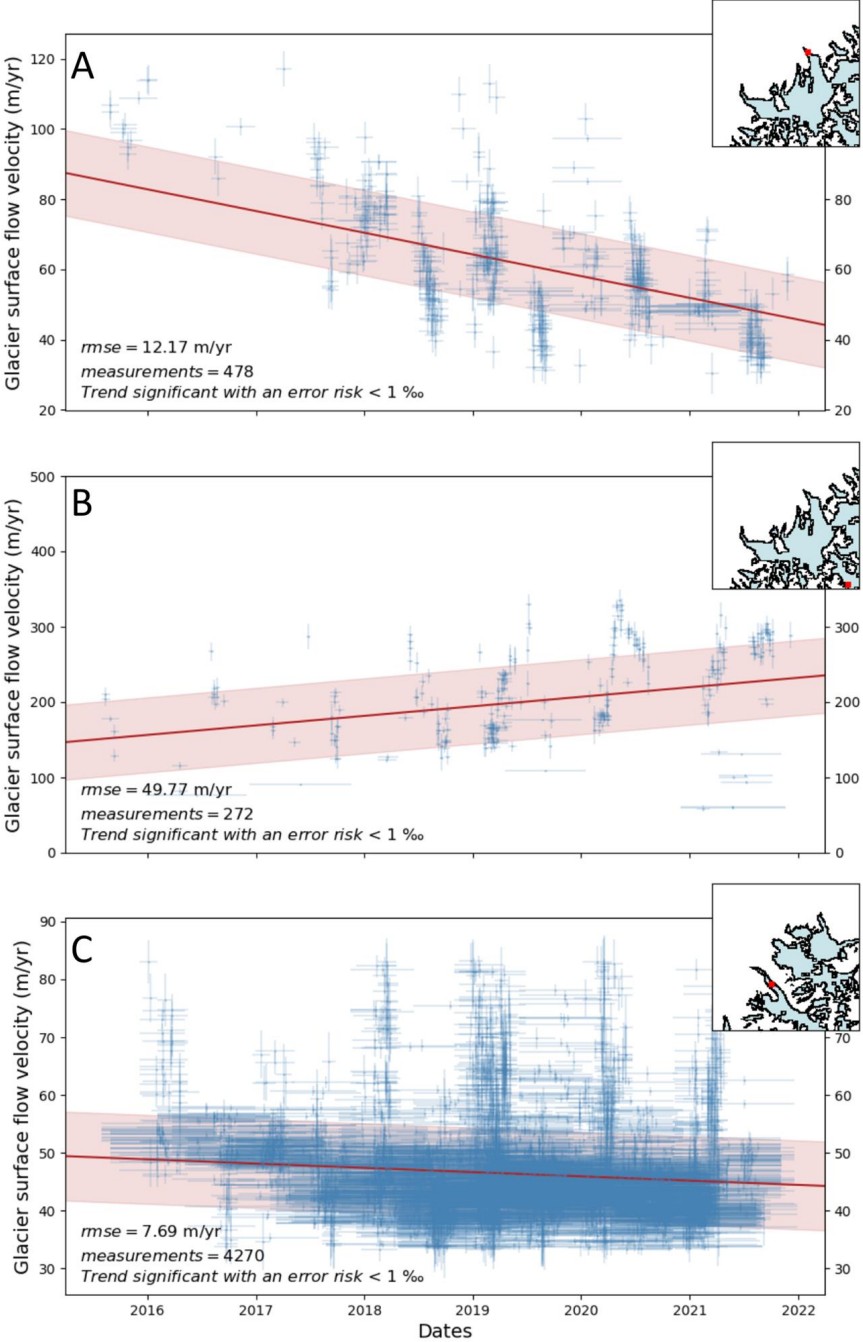

**Figure 6.** Glacier surface flow velocity data from Sentinel-2 for different locations of glaciers in the Mont Blanc massif. Individual measurements are plotted in blue, trends are shown in red and standard deviations around the trends are plotted in pale red: (**A**) velocities on Bossons Glacier, (**B**) velocities on Brenva Glacier, (**C**) velocities on the tongue of Argentière Glacier.

Figure 6 also shows that the density of data depends on the velocity. For the fastest areas of the glaciers, thanks to temporal decorrelation because there have been too many changes in the surface state, only the shorter temporal intervals of the image pairs can be used, thus limiting the number of potential image combinations. This is illustrated by the lower number of blue dots in Figure 6A,B and by the smaller horizontal bars on the blue dots that reflect shorter temporal offsets between the two images used for the correlation. A seasonal signal can be seen in the point clouds, particularly when velocities exceed 100 m/yr (i.e., Figure 6B), and only short temporal offsets give valuable results. On the contrary, when velocities are low (i.e., Figure 6C, around 50 m/yr), even if a seasonal increase in velocity can be seen in Figure 6C, it is partly hidden by the high density of points quantified from longer temporal intervals, which give lower velocities because they are aggregated over a longer period. Quantifying seasonal variations in glacier surface flow velocity is beyond the scope of this study, but our data set shows that Sentinel-2 data have good potential for studies in this direction.

Figure 7 shows the map of trends in glacier surface flow velocity over the period 2015–2021 for the Mont Blanc massif. The mask allows for distinguishing the pixels where the glacier surface velocity trend is significant.

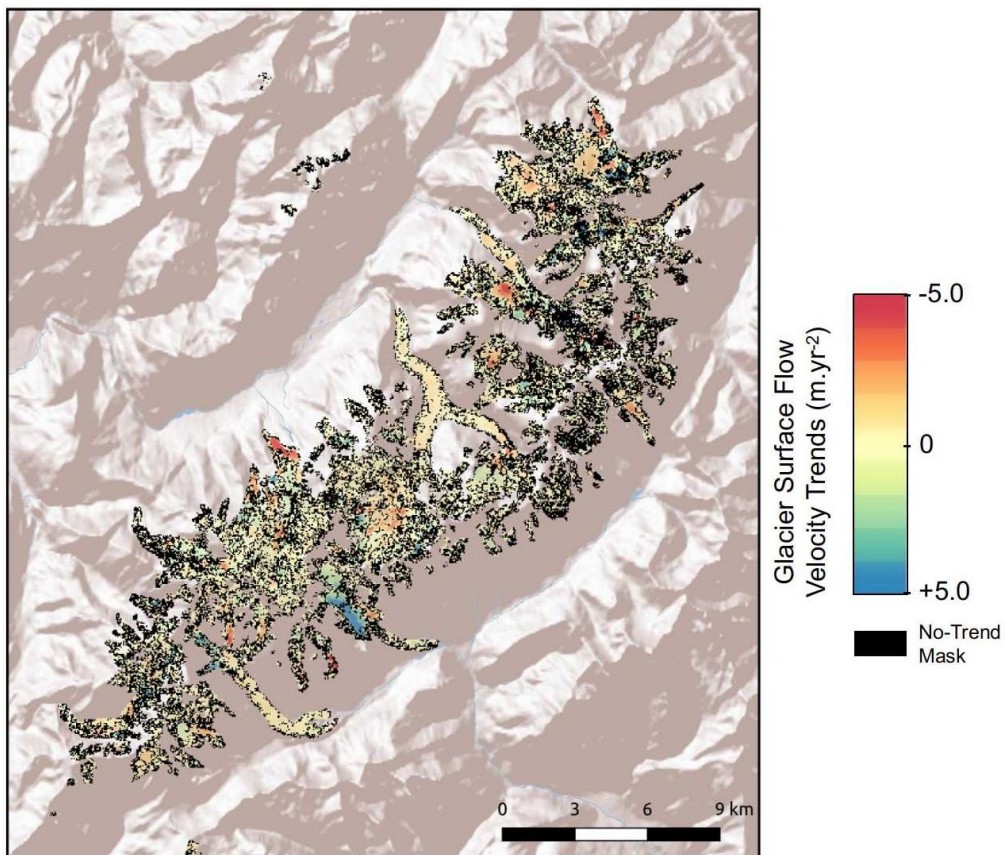

**Figure 7.** Map of trend in glacier flow velocity with a mask based on significance of the trend for the Mont Blanc test site.

Figure 8 illustrates the distribution of the pixels for which the trend is significant. The percentage of pixels for which the velocity trend is significant is about 32% of the total number pixels at the alpine-wide scale for which velocity data are available; the large majority of these pixels are in the lower reaches of the glaciers (i.e., the ablation areas). Figure 8 also shows that the overall trend represented by the median of the distribution is slightly negative ($-0.15$ m/yr$^2$). The lower and upper quartile values are $-0.73$ m/yr$^2$ and $+0.36$ m/yr$^2$, respectively. The largely observed negative trend is consistent with the glacier mass loss over the past decades and in particular over the period of interest (since 2015).

However, few areas appear to have a positive trend (i.e., acceleration). More focus on these areas is necessary to understand the processes involved in such changes; it could be related to a change in the thermal regime of the cold areas of polythermal glaciers.

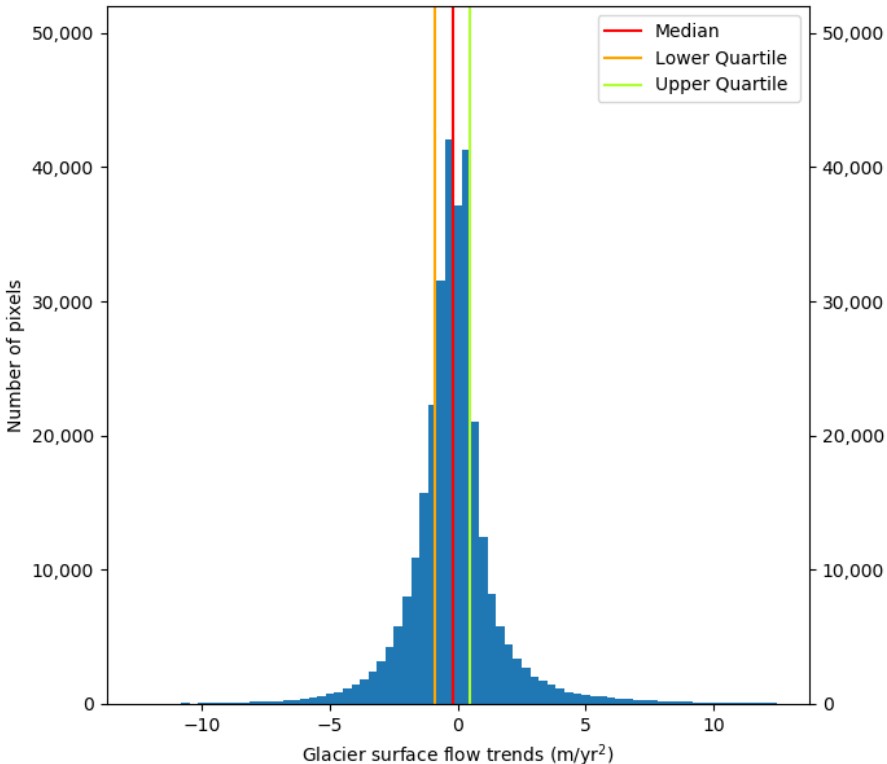

**Figure 8.** Histogram of the distribution of pixels where the trend in glacier surface flow velocity is significant (alpine range scale). Median, lower and upper quartiles are plotted in red, orange and green, respectively.

## 3. Methods

The overall workflow for data processing is shown in Figure 9. It consists of four modules: (1) database creation and image preparation; (2) glacier surface displacement calculation; (3) data calibration and geodatabase creation; and (4) postprocessing with data filtering and averaging to obtain glacier surface velocity maps.

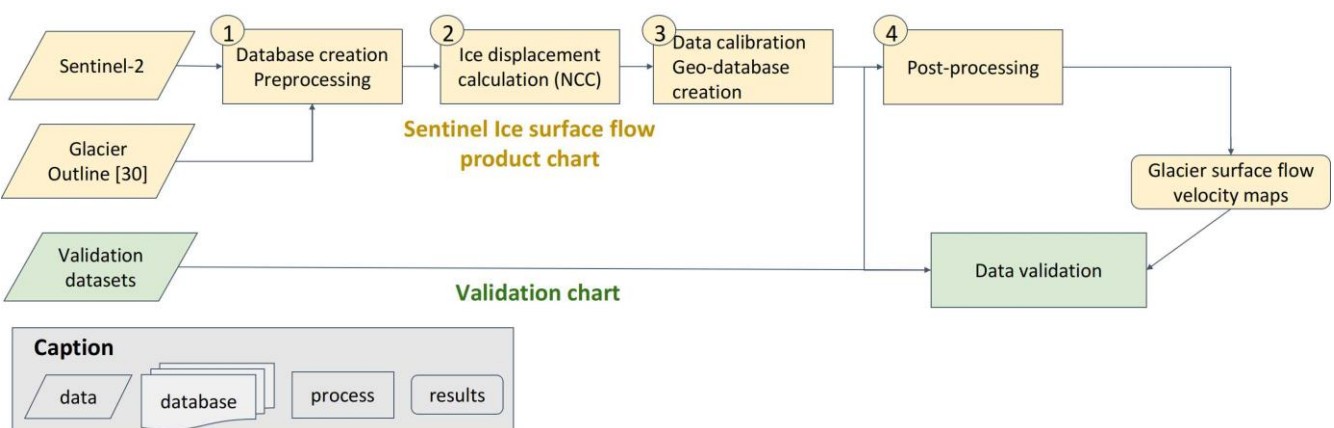

**Figure 9.** Overall chart of the workflow. Figures giving details on steps 1–4 can be found in their respective subsections.

All codes are written entirely in open-source programming languages: Python 3.7 and Fortran. The processing chain is designed for massively parallel processing with SQL (structured query language). A full description of the first three steps can be found in Millan et al. [10]; the last step is described in Mouginot et al. [14]. Hereafter, we provide only a synthesized description.

### 3.1. Step 1a: Database Initialization and Image Search

First, a shapefile with the outline of the study region is generated from the European Alps glacier inventory [25]. The perimeter includes both glaciers and stable ground off glaciers for calibration, but their extents are limited to reduce the computation time.

An SQL-MariaDB database is initialized in which a list of all existing images and associated metadata is stored for the studied region. Existing Sentinel-2 images are searched by using the API from ESA and Google. When the images are paired to map the displacement, another database is initialized. This second SQL-MariaDB database is fully integrated into the rest of the workflow and used to monitor and update the status of the processing step for each individual image pair. The use of an SQL-MariaDB database to update the processing steps allows the massively processing of image pairs in parallel, hence taking advantage of the full capacity of the HPC infrastructure.

### 3.2. Step 1b: Image Preparation and Preprocessing

The workflow of the preprocessing step is detailed in Figure 10. The image preparation module automatically downloads all available Sentinel-2 A and B images in the regions of interest. Sentinel-2 A and B data are provided as orthorectified images (L1C).

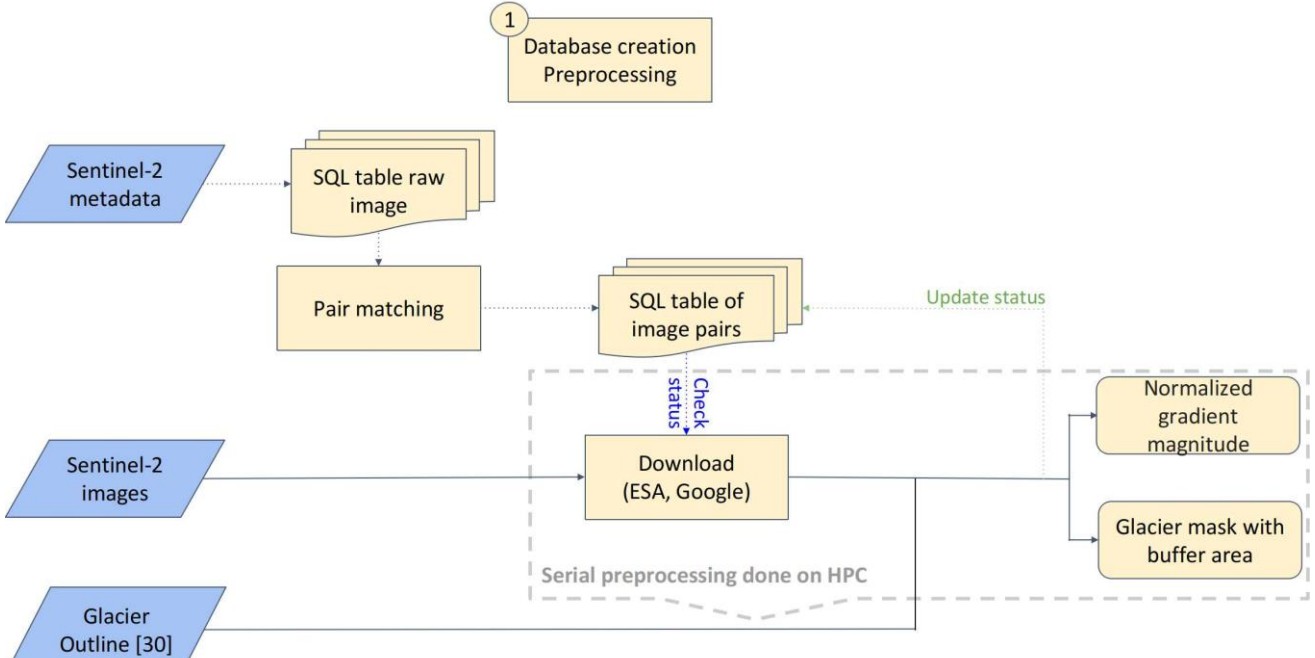

**Figure 10.** Workflow of the database creation and preprocessing, step 1.

In order to obtain accurate annually averaged maps of glacier surface flow velocity suitable for comprehensive regional-scale mapping, we form all the possible pairs of images with repeat cycles ranging from the sensor nominal cycle (5 days) to 400 days. In addition, only acquisitions made in the same orbits are matched, in order to minimize residual stereoscopic effects that may occur between adjacent orbit paths [26] that have different observation geometries. Finally, we apply a Sobel filter in the -x and -y directions to enhance the surface features, which seems to increase spatial coherence [9]. The result of this filter is stored as a complex binary file [9]. We use the near-infrared band #8 of Sentinel-2 (842 nm,

10-m resolution), which shows the best spatial resolution and low radiometric noise, and we therefore seem better situated for image offset tracking [27,28].

### 3.3. Step 2: Glacier Surface Displacement

The workflow of the image correlation step used to derive the glacier surface displacement is detailed in Figure 11. We modified the algorithm ampcor, which was part of the Repeated Orbit Interferometry Package (ROI-PAC) [29]. Our workflow uses the core of the cross-correlation algorithm written in Fortran-90. This algorithm is integrated into a Python environment to maintain both efficiency and flexibility with the rest of the workflow. The Python environment extracts the windows from the reference and secondary images and calls ampcor using the interface numpy.f2py. The Fortran-90 code of ampcor calculates a standardized cross-correlation map between the reference image window and the secondary image window. From this correlation map, a peak correlation value is determined. The two image windows are deramped and oversampled by a factor of two by using a 2D fast Fourier transform. The correlation procedure is then repeated around the previously determined peak value. The resulting new correlation map is itself oversampled by a factor of two, and the location of the correlation peak is then determined. Finally, the outputs are returned to the Python procedure.

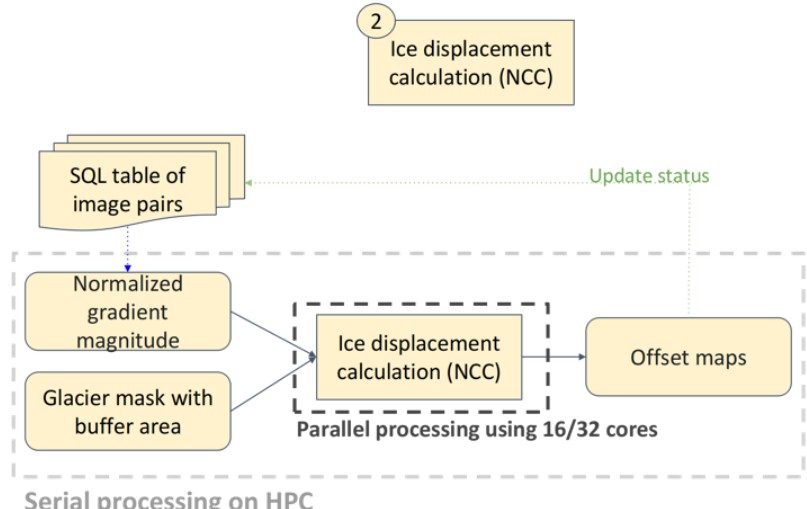

**Figure 11.** Workflow of the ice displacement calculation, step 2.

Thanks to the Python environment, new features can now be more easily implemented and added compared with using a Fortran-only code, such as the ability to restart a process in the event of a crash, the use of a mask to determine where to calculate the correlation or an initially guessed displacement map to guide cross correlation in a fast-moving region, thus reducing calculation time and avoiding correlation errors that are due to long-distance travel.

The mask provided to ampcor is derived from the shapefile of the study region delineated in step 1 of the workflow. It includes glaciers and surrounding stable ground areas for calibration purposes.

For the correlation, we used an adaptive search window ranging from 4 × 4 to 64 × 64, depending on the length of the repetition cycle, the resolution of the sensor and the average surface flow velocity of mountain glaciers according to previous studies and the tests that have been performed. Thus, we assume maximum surface velocities of about 1000 m/yr for the fastest parts of the glaciers in the Alps (e.g., ice fall of the Mer de Glace, Bossons Glacier).

According to a series of tests performed in Mouginot et al. [14], we used the optimized values of the sampling resolution (5 pixels) and of subimage/windows size (16 × 16 pixels). Because the window size is larger than the sampling resolution, each point on the offset map may not be completely independent of its direct neighbors. However, this approach

provides better results than what the interpolation of a coarser grid at a resolution of 50 m could provide. The size of the windows is appropriate for monitoring mountain glaciers because it allows for preserving characteristics on a small spatial scale.

### 3.4. Step 3a: Calibration of the Velocity Maps

Figure 12 presents the workflow for the calibration of the surface flow velocity data. Glacier surface velocity maps are automatically calibrated to correct potential biases due to relative geometric distortions or geolocation errors between two images. To do this, we mask the glaciers by using the glacier inventory of the European Alps [25,30] and calculate the average displacement on the stable ground areas [31]. This value is then subtracted from the entire displacement map. For Sentinel-2-derived maps, we used constant values because we did not observe a ramp, potentially because of unnoticed jitter in the displacement maps [10]. Therefore, we obtain zero-centered surface displacement fields in stable ground areas [32]. The average correction that is applied to Sentinel-2-derived data is 0.52 pixels, which approximately corresponds to the absolute geolocation specification (0.3 pixels) for multitemporal registration required by the Copernicus program [33]. It is important to note that such errors in the geolocation of images can lead to large errors when converted to glacier surface velocity. For example, uncalibrated glacier flow velocity maps with a 400-day repetition cycle would have a bias of 5 m/yr. On the other hand, errors in the velocities calculated with a 5-day repeat cycle could reach several hundred meters per year. The calibration of glacier displacement maps is therefore a key step in the processing of glacier flow velocities.

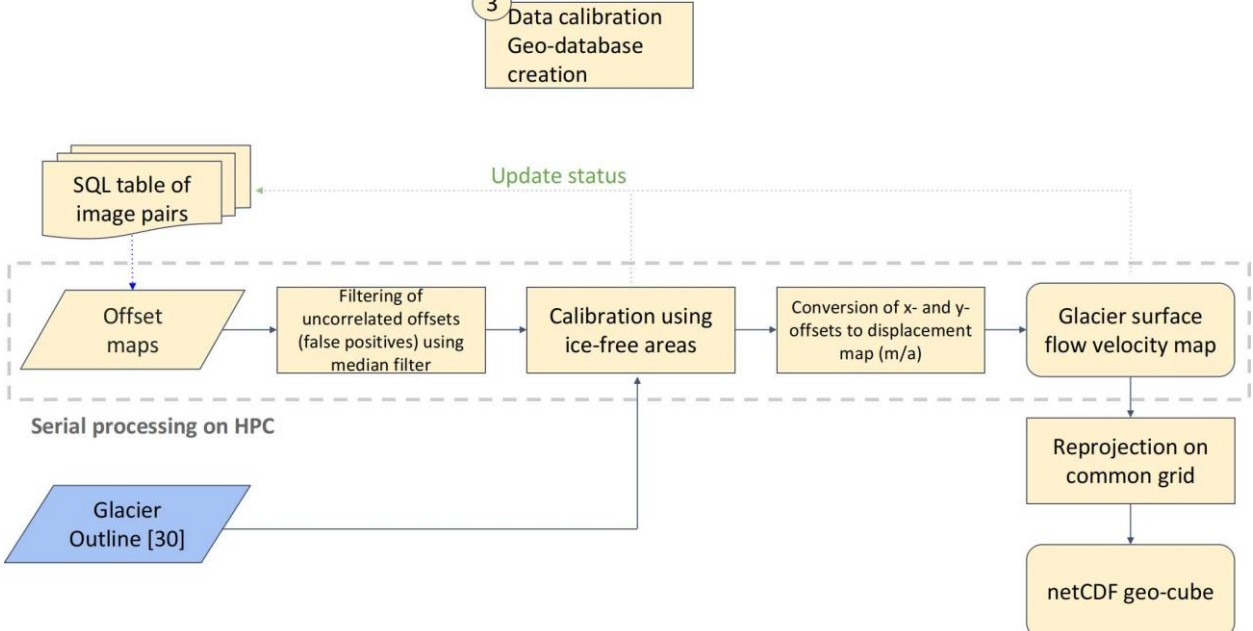

**Figure 12.** Workflow of the glacier surface flow velocity calibration and geodatabase creation, step 3.

We do not take into account the remaining stereographic effects, which we assume to be minimal for images taken in the same orbit, which is the case in our processing chain with Sentinel-2 data. Finally, the displacement maps are cleaned of outliers by using a median filter of 9 × 9 pixels [31]. As the median filter is used to detect outliers rather than to "smooth" the maps, there is no reduction in resolution. The displacement maps in pixels are then converted to glacier surface velocity in meters per year and geocoded using the Universal Transverse Mercator projection. The final calibrated maps are produced at a resolution of 50 m. The x- and y-components of the displacements in meters per year are

stored as GeoTiff. We use the GDAL library for all geographical transformations and the formatting of the final files.

### 3.5. Step 3b: netCDF Geocubes Database

To facilitate postprocessing, our region of interest is divided into areas of equal size, specifically 10 km × 10 km, with a resolution of 50 m (or 250 × 250 pixels). For each area, we extract and stack the velocity maps and associated metadata into "cubes" whose dimension z therefore corresponds to the number of calculated glacier surface flow velocity maps (Figures 12 and 13). The cubes overlap by 5 pixels, or 0.25 km, to avoid edge problems during postprocessing. This standardized data set where all maps are stored on a common grid can be easily used to extract time series of the surface flow velocity or calculate time-averaged maps. The cubes are stored in the GDAL-compatible netCDF format following the CF metadata conventions to allow for greater portability and facilitate distribution in the community. A cube file contains metadata about the cube itself (dimensions, corner coordinates, number of speed maps), surface displacement maps in meters per year in x/y directions, associated errors, projection information (UTM area, data, etc.), processing directory in our system, the dates of reference and secondary images, repeat cycles and sensor names. In total, 263 cubes are needed to cover the glaciers of the European Alps.

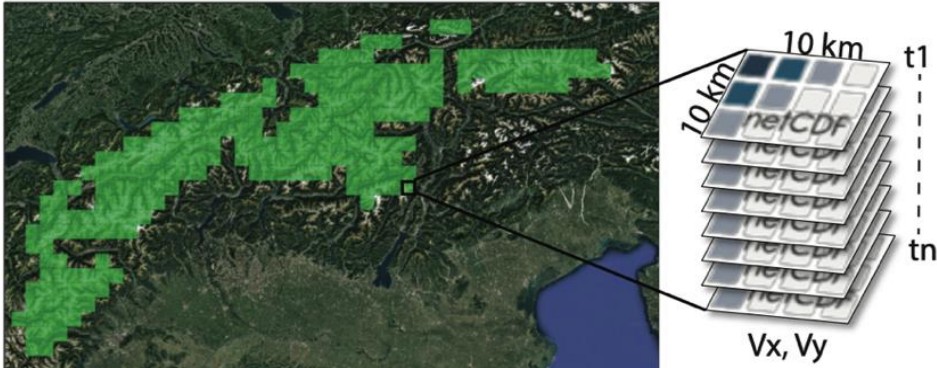

**Figure 13.** Data storage in 10 km × 10 km geocubes. Each layer of the cube (i.e., the vertical dimension) contains the calculated glacier surface flow velocity maps for every pair of images that has been processed.

### 3.6. Step 4. Postprocessing

Figure 14 presents the workflow of the postprocessing steps to generate glacier surface flow velocity products.

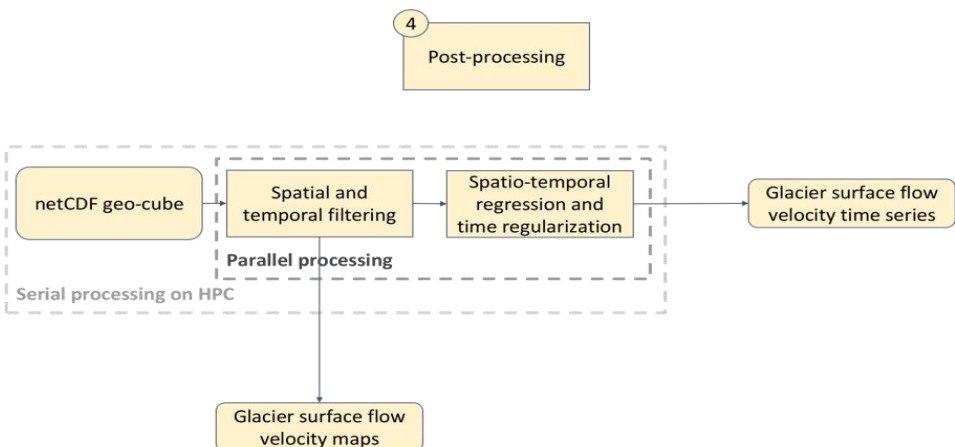

**Figure 14.** Workflow of the postprocessing, step 4.

Time-averaged glacier surface velocity maps are computed on the basis of the data contained in the geocubes described above. Several geocubes can be simultaneously and independently processed, thus optimizing the time needed to postprocess all the Alps. For each geocube, a temporal pixel-by-pixel filtering is first performed. Next, the filtered maps are aggregated into annual mosaics, which are spatially filtered. The local results obtained for each geocube are finally merged to obtain comprehensive annual maps covering the entire Alps.

For the data filtering, we have relied on the tests performed by Mouginot et al. [14]. The filtering stage includes the following filters:

- Filter out obvious outliers, i.e., nonplausible values for mountain glaciers of the alpine type. In the Alps, the fastest glaciers are flowing at a pace of up to 1000 m/yr [10,11,34]. Therefore, surface flow velocities faster than 1000 m/yr can be discarded.
- Apply a median spatial filtering. We eliminate pixels for which the pixel displacement deviates by more than three units from the $9 \times 9$ pixels' median value [10,31]. This filtering is separately applied on the two components of speed ($v_x$ and $v_y$). When one component is filtered out, the other component is also removed.
- Filter out the temporal baselines (time between two acquisitions of a pair of images used to estimate displacement) on the basis of the glacier flow magnitude vs. the expected accuracy. Indeed, the noise associated with the measured displacements increases inversely with the temporal baseline [10]. Thus, the noise associated with short cycles makes it impossible to measure the shortest displacements. Conversely, the longest temporal baselines do not capture the fastest displacements.
- Filter out the flow direction as it is unexpected that the flow direction significantly varies over the study period from 2015 to 2021. We test different angles for which the direction of the velocity vector is allowed to vary around the median flow direction. Angle values from $\pm 45°$ have been tested. Each displacement value whose direction is outside the tested range is discarded.
- Filter out the distribution of the surface flow velocities. For each pixel, we consider the distribution of the surface flow velocities quantified over the entire study period (2015–2021). We then eliminate the distribution tails for the percentile values below 20% and above 80%. Thus, 40% of the extreme values of the distribution have been filtered out.

The outputs after applying all the filters have been evaluated by comparing them with in situ d-GNSS measurements. The root-mean-square error (RMSE) is 11.6 m/yr and the coefficient of determination ($r^2$) is 0.9 [14].

To aggregate the individual velocity measurements in order to create the most accurate and most complete annual glacier flow mapping in the Alps, four simple and robust aggregation methods have been tested by Mouginot et al. [14]: (1) the median of individual velocity measurements for each year from 2015 to 2021; (2) the weighted average for each year from 2015 to 2021; (3) the OLS regression between 2015 and 2021; and (4) the Theil–Sen estimator, which is based on the fit of a linear regression in a scatter plot that factors in the median of the slopes of all the possible lines passing through pairs of points. They showed that the performances of the different aggregation methods are very close when comparing the results with in situ d-GNSS data, even if the OLS regression had a slightly better RMSE (10.5 m/yr) and an $r^2$ of 0.92. Therefore, this aggregation method has been used to produce the data set. Finally, a spatial filtering on the trend map that uses either a $3 \times 3$ Gaussian or a median filter is applied.

**Author Contributions:** A.R. and J.M. conceptualized the study. J.M. and R.M. developed the workflow for data processing. J.M., A.R. and E.D. developed the methodology for data reduction. J.M. and E.D. processed the satellite data. A.R., J.M. and E.D. validated the products and analyzed the results. All coauthors contributed to the discussion. A.R., E.D. and R.M. wrote the original draft and reviewed the manuscript. All authors have read and agreed to the published version of the manuscript.

**Funding:** This research was funded by the European Space Agency (ESA), specifically its "Glacier Science in the Alps" project, coordinated by Frank Paul (University of Zurich), contract no. 4000133436/ 20/I-NB. This work has also been supported by a grant from LabEx OSUG@2020 (*Investissements d'Avenir*—ANR10 LABX56) and the programs MaISON (funded by the French space agency: CNES) and SOSice (funded by the French research agency: ANR-19-CE01-0011-01).

**Institutional Review Board Statement:** Not applicable.

**Informed Consent Statement:** Not applicable.

**Data Availability Statement:** Sentinel-2 images used in this study to generate the glacier surface velocity maps are available through the API from ESA and Google.

**Acknowledgments:** In situ measurements of glacier surface flow velocity used to validate the satellite-derived products were provided by the French Service National d'Observation GLACIOCLIM (https://glacioclim.osug.fr/, Univ. Grenoble Alpes—OSUG, CNRS—INSU, IRD, IPEV, INRAe). We are grateful to Steven Plummer (ESA ESRIN, technical officer of the "Glacier Science in the Alps" project) and Frank Paul for fruitful discussions and comments on the manuscript. We thank the OSUG-Data Center (V. Chaffard, C. Coussot, J. Schaeffer, B. Boutherin and F. Malbet) for its support in the distribution of the glacier surface flow velocity product.

**Conflicts of Interest:** The authors declare no conflict of interest.

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
