# Peer review of "Satellite-Derived Annual Glacier Surface Flow Velocity Products for the European Alps, 2015–2021"

_data, 2015_

Round 1

Reviewer 1 Report

With the free-open sentienel-2 optical images, the pixel offset correlation method has been widely used to derive a surface deformation with high spatial-temporal resolution. This work plans to provide a public dataset product of glacier surface flow velocity for the European Alps with the Sentinel-2 images. I think it is a valuable work for the detection of local hazards associated with glacier destabilization, and its present statement can be accepted. There are several main comments should be paid attention as follows.

1. In order to friendly access for users, a detailed summarized information for this product should be included in a table, such as its key parameters, characteristics, comparisons with other dataset products.

2. Note that in-situ d-GNSS observations have been used to evaluate the precision of the product. Please add these d-GNSS sites on the figure, and add a detailed evaluation strategy for its precision.

3. A minor suggestion, some value close to zero are plotted with black (or dark grey) (such as Figure 2, 3, 5, 7), please change them with more clear color.

Reviewer 2 Report

Please see the attached file in Word

Reviewer 3 Report

This is a paper introducing the European Alps glacier ice velocity product produced by the authors. From the paper, the ice velocity product has excellent performance. It is valuable data for the study of European Alps glacier changes. The paper also has a detailed description of methods and product. But I think this paper can be improved in the following aspects:

1) The abstract needs to be rewritten. It lacks relevant introduction about product data source, generation method and product performance.

2) The paper lacks the conclusion. Even if it is a data descriptor, it also needs a systematic summary.

3) The introduction of Figure 3 is very unclear. I don't understand what each sub-figure of Figure 3 is.

4) Please redraw Figure 3. The place names marked in Figure 3 are completely unclear. I don't know why the author gave them.

5) The trend of surface ice flow velocity should be acceleration, right? Its unit should be m/yr/yr, or m/yr ^ 2? But it should not be m/yr.

6) Since the author mentioned ”Figure 1 shows the study region together with the three test sites where Sentinel-2 derived surface flow velocity data have been compared with in situ d-GNSS data for validation purpose”, why is there no verification result of surface ice flow velocity products made by authors using these data? The cross evaluation of external data is very important for the estimation of product uncertainty.
